# Molecular Relay Stations in Membrane Nanotubes: IRSp53 Involved in Actin-Based Force Generation

**DOI:** 10.3390/ijms241713112

**Published:** 2023-08-23

**Authors:** Tamás Madarász, Brigitta Brunner, Henriett Halász, Elek Telek, János Matkó, Miklós Nyitrai, Edina Szabó-Meleg

**Affiliations:** 1Department of Biophysics, Medical School, University of Pécs, H-7624 Pécs, Hungary; 2Institute of Biology, Faculty of Sciences, University of Pécs, H-7624 Pécs, Hungary; 3Department of Immunology, Faculty of Science, Eötvös Loránd University, H-1117 Budapest, Hungary

**Keywords:** actin, fluorescence microscopy, IRSp53, membrane nanotube, protein–protein interactions

## Abstract

Membrane nanotubes are cell protrusions that grow to tens of micrometres and functionally connect cells. Actin filaments are semi-flexible polymers, and their polymerisation provides force for the formation and growth of membrane nanotubes. The molecular bases for the provision of appropriate force through such long distances are not yet clear. Actin filament bundles are likely involved in these processes; however, even actin bundles weaken when growing over long distances, and there must be a mechanism for their regeneration along the nanotubes. We investigated the possibility of the formation of periodic molecular relay stations along membrane nanotubes by describing the interactions of actin with full-length IRSp53 protein and its N-terminal I-BAR domain. We concluded that I-BAR is involved in the early phase of the formation of cell projections, while IRSp53 is also important for the elongation of protrusions. Considering that IRSp53 binds to the membrane along the nanotubes and nucleates actin polymerisation, we propose that, in membrane nanotubes, IRSp53 establishes molecular relay stations for actin polymerisation and, as a result, supports the generation of force required for the growth of nanotubes.

## 1. Introduction

Communication with the environment is essential in all living biological systems. Cells form heterogeneous structures to facilitate cell-to-cell communication and material transport. Cell protrusions play a key role in the former. They are capable of dynamic movements such as lateral motions, fast elongation (linear growth) and retrograde flow (withdrawal).

Actin is known as a key regulator in the generation of membrane invaginations and cell extensions. Actin-associated membrane-deforming (sculpturing) proteins also play a primary role in the cell membrane rearrangement, since they directly alter the shape of the plasma membrane and the organisation of lipids. The formation of cell protrusions is a complex process regulated by actin polymerisation and actin-associated scaffolding proteins [1,2,3].

The inverse Bin-Amphiphysin-Rvs (I-BAR) sculpturing domain—a member of the highly conserved BAR domain superfamily proteins—is a basic mediator of the remodelling of eukaryotes’ cytoplasm membrane. The I-BAR domain was originally identified as the N-terminal region of the missing-in-metastasis (MIM) and insulin receptor tyrosine kinase substrate 53 (IRSp53) proteins. I-BAR exists as a six-helix bundle dimer and contains positively charged amino acids on its ends that provide binding sites for the plasma membrane and for the actin filament [4,5,6,7]. I-BAR domain binds the phosphatidyl-rich membrane regions and is capable of deforming them into a tubular structure [5,7,8], facilitating the development of protrusion-like structures (such as filopodia) by inducing the formation of negative membrane curvatures. During membrane curvature formation, I-BAR domains influence the dynamics of lipid organisation and induce a strong, microscopic-scale aggregation of phosphatidylinositol 4,5-bisphosphate (PI(4,5)P2) phospholipids [8]. The IRSp53 C-terminal part contains a WASP homology 2 (WH2) domain that can bind the monomeric form of actin [9]. The SRC homology 3 (SH3) domain of IRSp53 binds to the proline-rich regions of several proteins regulating the actin dynamics (such as WAVE-2, Mena and Eps8) [10,11,12], and the CRIB domain interacts with the small GTPase protein Cdc42; consequently, the full-length IRSp53 affects not only the shaping of the membrane curvature but also the actin polymerisation. IRSp53 expression has been reported in several mammalian tissues and cell types, particularly in neurons. Certain alleles of IRSp53 are associated with neurological diseases, such as changes in the number of dendritic spines, attention deficit disorders, and hyperactivity. Furthermore, IRSp53-knockout mice exhibited deficiencies in learning, memory, and synaptic plasticity [13].

Membrane nanotubes (NTs) are filopodium-like intercellular bridges that provide a physical connection even between distant cells [14]. Based on in vitro studies conducted on cell cultures, filopodia are attached to the surface of the culture dish, while NTs float freely in the cell culture medium [15]. In vitro experiments revealed that membrane nanotubes are involved in various transport processes—for instance, mediating the transfer of calcium ions, cell organelles, lipids, numerous proteins (including misfolded ones, such as amyloid β, tau, and prions), nucleic acids, and immune-costimulatory molecules, and their role was reported in the effective propagation of bacteria and medically relevant viruses among cells [14,16,17,18,19,20,21]. NTs are enriched in cytoskeletal filaments. Actin not only serves as the skeleton of NTs but can mediate some transport processes within these nanoscale structures [16,22], and similarly to filopodia, microfilaments were proven to be essential in the development of the NTs [17]. Despite the growing evidence regarding their physiological functions, the molecular mechanism of NT growth is not yet clearly understood. Until now, two different strategies have been suggested by which different cell types manage to form NTs [20,23]. One means of NT formation is when a filopodium-like protrusion is formed and attached to a nearby cell (e.g., in PC12 cells, dendritic cells, and neural crest cells). The other possibility is when two cells in physical contact move away from one another and remain connected with the help of NTs (e.g., in T and B lymphocytes) [24,25,26,27,28].

We have already published evidence that the lipid composition of the plasma membrane affects NT formation, and NT growth shows a linear dependency on the levels of gangliosides in the membrane. Several constituents of lipid rafts in the inner leaflet of the cell membrane, such as phosphatidylcholines and sphingomyelins, aid in the bending of the membrane, contributing to the formation of negative membrane curvature [29]. Beyond these molecular conditions, numerous mechanical factors affect the process of tubulation [30]. For instance, microtubule-based motor proteins can exert the necessary force for tube formation on the membrane by pulling a tube during their movement on the microtubule. Membrane tension may also contribute to the formation of tube-like projections, e.g., due to the steric effects of accumulated proteins in the lipid bilayer [1,30,31].

To address these issues, in the present work, we explored the effects of the full-length IRSp53 protein and its N-terminal I-BAR domain on the formation and morphology of filopodia and membrane nanotubes, as well as on individual actin filaments, by live-cell confocal laser scanning microscopy (LC-CLSM), total internal reflection fluorescence (TIRF) microscopy, and actin polymerisation assays. The measurement of COS-7 mammalian kidney and A20 B lymphocyte cell lines revealed that both the full-length protein and its N-terminal domain induce the growth of filopodia independently of the cell type; however, their effects on the number of NTs are strongly dependent on how the given cell grows the tube (cell dislodgement/making contact). Furthermore, the characteristic morphological changes in NTs and filopodia suggest that both the full-length IRSp53 and its N-terminal I-BAR domain influence actin assembly by promoting both the nucleation and capping of actin, as confirmed by our TIRFM experiments. Here, we demonstrate that IRSp53 and even the I-BAR domain itself have the ability to promote actin assembly by inducing the formation of actin seeds. 

## 2. Results

### 2.1. IRSp53 Is Expressed in the NTs of COS-7 Kidney and A20 B Lymphoma Cells

Using confocal laser scanning microscopy, we observed that the African green monkey kidney COS-7 cell line and A20 mouse B cell line of reticulum cell sarcoma showed endogenous IRSp53 expression in their NTs (Figure 1). IRSp53 was distributed within the entire cell but the cytoplasm under the membrane was particularly enriched. 

IRSp53 showed colocalisation with filamentous actin under the membrane, in the cytoplasm and in the NTs (Figure 2 and Figure 3). As the signal intensities of the fluorophores applied for visualisation differed strongly, Manders’ colocalisation coefficients (M1 and M2) were used to assess the degree of colocalisation of IRSp53 with actin (Figure 2J,T and Figure 3J,O). Colocalisation coefficients in COS-7 cells under the membrane and along the stress fibres were very similar. The M1 (IRSp53) colocalisation coefficient was 0.99, while for actin (M2) it was 0.53. The results suggest that nearly all (99%) of the IRSp53 (green pixels) colocalised with actin filaments (red pixels), while approximately half (53%) of the actin filaments colocalised with IRSp53. In the NT, the measured M1 colocalisation coefficient (IRSp53) was lower (90%) and M2 (actin) was slightly higher (66%). In the other cell type (A20 B mature cells), stress fibres could not be observed. The colocalisation coefficients were the following: M1 (IRSp53) 52% (under the membrane) and 93% (in the NTs); M2 (actin) 83% (under the membrane), and 92% (in the NT), respectively. It is interesting to note that, in COS-7 cells, nearly all IRSp53 colocalised with actin and only half of the actin was colocalised with IRSp53, while in A20 cells an opposite colocalisation distribution was observed. 

Cells transfected with LifeAct-GFP were used as a control in live-cell microscopy experiments, since LifeAct-GFP expression affected neither the number of filopodia nor the NT growth frequency in COS-7 and A20 cells. Both mCherry-I-BAR and mCherry-IRSp53 were observed not only in the cytoplasm of the examined cell lines but also in the studied cell protrusions. Despite the fact the overexpression of I-BAR or IRSp53 strongly influenced the morphology of the cells (see below), control experiments showed that actin filaments were mostly present in the cell protrusions (Figure 4). Furthermore, I-BAR, or IRSp53 overexpression did not influence the functionality of the examined NTs (Appendix A). 

### 2.2. Overexpression of IRSp53 Protein and Its I-BAR Domain in COS-7 Kidney Cells

To quantify the effect of the I-BAR domain and IRSp53 on the morphology of the cells, we characterised both the filopodia and the NTs formed following the overexpression of these proteins. The results obtained with COS-7 cells are summarised in Table 1. In the control cultures, i.e., in non-transfected or in LifeAct-GFP transfected cells, an average of approximately two filopodia in each cell were observed. The overexpression of either the I-BAR domain or IRSp53 protein resulted in a significant increase in the number of filopodia (Figure 5C,D,G,H,K,L and Figure 6D), and this effect was somewhat greater in the case of the I-BAR domain. Parallel to the increase in the number of filopodia, their lengths also increased. The filopodia were approximately two times longer in the case of the I-BAR and four times longer regarding the IRSp53 (Table 1). These results indicate that the effect of the full-length protein on the elongation of the filopodia was more pronounced. Similar to these observations, the elevated level of I-BAR or IRSp53 increased the number and frequency of NTs by nearly two-fold (Figure 6A). At the same time, the average diameter of the NTs decreased from 1.06 ± 0.03 µm (observed for the control cells) to 0.56 ± 0.01 µm for I-BAR and 0.67 ± 0.02 µm for IRSp53 transfected cells. The length of the formed NTs was 21.50 ± 1.13 µm in control cells, and remained similar after I-BAR overexpression (20.25 ± 0.77 µm). However, the overexpression of IRSp53 resulted in an increase in the NT length (36.13 ± 1.55 µm) (Figure 6E,F), coincidently with the observations made with filopodia.

**Table 1 ijms-24-13112-t001:** The results of the analyses obtained with COS-7 cells and their transgenic modifications.

Parameter	COS-7 Control	I-BAR Overexpression	IRSp53 Overexpression
Number of filopodia per cell	2	62 ± 4	38 ± 3
Average length of filopodia (µm)	1.80 ± 0.13	3.90 ± 0.19	8.15 ± 0.33
Relative frequency of NTs	0.29 ± 0.02	0.75 ± 0.03	0.54 ± 0.03
Average diameter of the NTs (µm)	1.06 ± 0.03	0.56 ± 0.01	0.67 ± 0.02
Length of the NTs (µm)	21.50 ± 1.13	20.25 ± 0.77	36.13 ± 1.55
Relative branching	0.33 ± 0.11	0.77 ± 0.12	0.78 ± 0.06
Length of the branches (µm)	4.80 ± 0.42	5.82 ± 0.34	8.77 ± 0.50
Accumulation level in the NTs	0.22 ± 0.05	0.50 ± 0.07	0.50 ± 0.08

When further investigating the images, we also observed that the branching activity of the membrane nanotubes increased significantly when either IRSp53 or I-BAR was overexpressed (Figure 5F,J and Figure 6C). An average of 0.33 ± 0.11 relative branching was observed in NTs in control cells. Following overexpressing IRSp53 or its I-BAR domain, the number of relative branching increased to 0.77 ± 0.12 and 0.78 ± 0.06, respectively. Similarly to the length of the NTs, the length of the branches only changed significantly following IRSp53 transfection (Table 1 and Figure 6G). When I-BAR and IRSp53 were visualised in the transfected cells, their accumulation in the membrane nanotubes was also apparent, as reflected by the measured protein accumulation levels (Table 1 and Figure 6B). Line scan analysis along the NTs was used to verify the presence of protein accumulation (Appendix A).

### 2.3. Overexpression of IRSp53 Protein and Its I-BAR Domain in A20 B Lymphoma Cells

The microscopic data obtained with A20 B cells are summarised in Table 2. LifeAct-GFP transfected control A20 cells formed an average of 4.3 ± 0.5 filopodia; the growth frequency of filopodial projections was significantly higher following mCherry-I-BAR (16.30 ± 2.12) and mCherry-IRSp53 (13.8 ± 1.6) transfection. I-BAR had a somewhat more significant effect on the ability of A20 cells to form filopodia than IRSp53, as in the case of COS-7 cells. The length of the filopodia increased from 1.24 ± 0.06 µm (in control cells) to 3.05 ± 0.16 µm and 2.18 ± 0.11 µm at higher expression levels of I-BAR or IRSp53, respectively (Figure 7 and Figure 8). In the formation of NTs, differences were observed compared to COS-7 cells (Figure 7 and Figure 8). Regarding control cells the relative NT frequency was 0.16 ± 0.03, while the value of this parameter was 0.12 ± 0.03 and 0.19 ± 0.04 following the overexpression of I-BAR or IRSp53, respectively. The NT length in the control cells was 26.33 ± 5.59 µm, and the overexpression of I-BAR (21.36 ± 2.29 µm) or IRSp53 (31.75 ± 4.17 µm) did not significantly alter the value of this parameter. However, similarly to COS-7 cells, IRSp53 overexpression caused the formation of the longest tubes in A20 cells. Regarding the thickness of NTs, the I-BAR (0.93 ± 0.12 µm) and IRSp53 (1.24 ± 0.08 µm) had relatively few effects (control: 1.35 ± 0.10 µm). The overexpression of either protein enhanced the NT branching. The value of this parameter was 0.12 ± 0.80 in control cells and increased following the overexpression of I-BAR (0.55 ± 0.19), increasing even more substantially following the overexpression of IRSp53 (1.00 ± 0.31). Unfortunately, the length of these branches could not be statistically analysed due to their low frequency. The accumulation puncta of I-BAR and IRSp53 in transfected cells were not pronounced (control: 0 ± 0; I-BAR: 0.27 ± 0.16 and IRSp53: 0.36 ± 0.16) (Figure 8). 

A summary of the effect of the overexpression of I-BAR and IRSp53 on the NTs and filopodia of the examined cell types is depicted in Appendix A.

### 2.4. The Effects of IRSp53 and Its I-BAR Domain on Actin Polymerisation

The possible effects of different concentrations of the I-BAR domain and the full-length IRSp53 protein on the assembly of individual actin filaments were investigated under in vitro conditions using recombinant I-BAR and IRSp53 proteins by TIRF microscopy and pyrene-labelled actin-based polymerisation assays (Figure 9 and Figure 10). The number of actin filaments generated in TIRFM experiments in the presence of I-BAR was counted. The TIRF micrographs of the assembly of actin with IRSp53 were not as obvious; therefore, in these cases, the size distribution calculated from the area of the actin filaments was analysed. Both sub- and superstoichiometric conditions were examined. In the absence of I-BAR, 85 ± 55 filaments were counted and when the actin:I-BAR concentration ratio was 10:1, the number of observed filaments was 100 ± 53, indicating that, at very low concentrations, the I-BAR had little effect on the number of assembled actin filaments; however, the actin filaments were slightly longer (15.67 ± 4.27 µm (0.5 μM actin /control/), 19.08 ± 5.0 µm (actin:I-BAR = 10:1)) (Figure 9A,B,F). At an elevated yet still substoichiometric I-BAR concentration (actin:I-BAR = 5:1), the number of filaments decreased to 22 ± 14 (Figure 9E). In the presence of a higher concentration, 60 µM of I-BAR (which corresponds to actin:I-BAR = 1:120), the number of actin filaments dramatically increased to 450 ± 150 (Figure 9D,E) and the length of the filaments decreased (Figure 9C,F). At this high concentration, the effect of I-BAR showed saturation, and long actin filaments were not observed (Figure 9D).

The effect of the I-BAR domain on the actin assembly was also investigated in fluorescence spectroscopic assays using 2.5 μM of pyrene-labelled actin (Figure 9G). We observed an increase in the pyrenyl intensity at the early phase of actin assembly in the presence of relatively high concentration of I-BAR (actin:I-BAR = 1:36), suggesting that superstoichiometric amounts of I-BAR can promote the formation of actin nuclei. The kinetics of substoichiometric concentration of I-BAR did not differ from the pyrenyl trace of the 2.5 µM actin used as a control.

The effects of sub- and superstoichiometric concentrations of IRSp53 on actin filaments were also analysed based on TIRF micrographs. In the absence of IRSp53, both shorter and longer filaments were observed. At substoichiometric concentrations of IRSp53, the distribution of filaments’ length shifted toward the medium-length filaments, while at superstoichiometric concentrations, mainly short filaments were detected (Figure 10).

Our microscopic and spectroscopic data demonstrate that, at superstoichiometric concentrations, the I-BAR and IRSp53 induced similar changes in the length distributions of actin filaments. These observations also showed that both proteins can nucleate actin.

## 3. Discussion

### 3.1. Background and Aim

The relationship between distant cells is often achieved by cell protrusions appearing on the surface of the cells. Actin polymerisation is thought to be responsible for the formation of many of these cell extensions (lamellipodia, filopodia, membrane ruffles and nanotubes). The tube-like deformation of the membrane is also dominantly directed by I-BAR domain proteins attached to the inner surface of the membrane tubule [32]. Since these bridges are involved in the progression of several pathological conditions (e.g., cancer, viral or bacterial infections), they provided an exciting target for biomedical research in the last 15 years; nevertheless, several key questions in the field are still unanswered. There is a consensus that actin plays a central role in the initial formation of NTs [14,17,33]. Considering the flexible nature of actin filaments, actin’s ability to support the growth of long membrane nanotubes requires further explanation. One interesting aspect is how the long actin filaments can maintain their mechanical support while producing force. Even considering their ability to form bundles, the number of participating actin filaments would decrease along the NTs, since not all of them can progress together with the growing actin bundle [34]. A possible mechanism for this process should include molecular events through which the number of actin filaments is kept constant, even when the growth in some of the participating polymers stops and, at the same time, the growing and pushing actin bundle can find new mechanical supporting points along the NTs. The extreme mechanical stability of the relatively long lymphocyte TNTs [29] also underlines the importance of this question.

Thus, the aim of this work was to investigate the possibility of the formation of mechanical support points along the membrane nanotubes for the growing actin filament bundles. They are referred to here as ‘molecular relay stations’. These protein clusters would serve as the structural framework for the force production of actin filaments over long distances. Considering their envisaged function, the protein serving as the basis of the stations needs to fulfil several key criteria. It has to be endogenously present in living cells and, more importantly, in their membrane protrusions. Additionally, these proteins must be able to bind to the membrane of cells to establish the localisations and should facilitate the polymerisation of actin to provide new filaments for the actin bundles. Once these criteria are fulfilled, one can conjecture that the increased concentration of this protein results in increases in the number and length of membrane protrusions.

### 3.2. Key Observations

Due to its known functional properties, we studied the interactions of IRSp53 full-length proteins and its membrane-binding I-BAR domain with actin. To provide a broader understanding, the experiments were carried out on two different cell types. Despite the observation that the overexpressed proteins in certain cases acted slightly differently on the examined cell types, their effects were nearly the same in multiple aspects. Data are publicly available regarding the endogenous expression of IRSp53 in the cell body of monkey kidney [35,36] and mouse B lymphocytes [37]. We showed that IRSp53 is endogenously expressed in the NTs of COS-7 monkey kidney and A20 mouse B lymphoma cells; furthermore, it is localised to the membrane NTs under control or overexpressed conditions (Figure 5 and Figure 7). When IRSp53 or its I-BAR domain was overexpressed in COS-7 or A20 cells, the number and length of filopodia increased, but only the full-length protein was able to cause substantially similar changes in the NTs of both cell lines to those induced in filopodia. These results showed that IRSp53 and its I-BAR domain play an important role in the formation and elongation of membrane nanotubes. We also observed a decrease in the thickness of the nanotubes, which was likely attributed to their faster growth. In the presence of elevated IRSp53 or I-BAR concentrations, the branching activity of the membrane nanotubes also increased in correlation with the membrane-sculpturing function of the I-BAR domain. Additionally, regarding the results found in our TIRFM and actin polymerisation experiments (Figure 9 and Figure 10), we demonstrated that IRSp53 or I-BAR can promote the formation of actin nuclei.

### 3.3. General Concept

Due to the similarity of the structures of filopodia and NTs, and the known overlaps between the proteins participating in their function, one would envisage that, to a certain extent, these structures are regulated by similar molecular mechanisms. However, it was published that identical signalling mechanisms mediated by certain actin regulatory complexes may have contrary effects on the formation of filopodia and NTs, e.g., Cdc42, VASP and IRSp53 can collaborate with one another to inhibit NT growth and, at the same time, promote the inception of filopodia, at least in neurons [15], suggesting that filopodia and NTs can behave as unrelated cell extensions. At the beginning of their growth, they appear similar, especially in those cases when the NT is formed through filopodial interplay, suggesting that the same set of proteins is responsible for the initiation of their formation. However, at a defined point in their growth, a decision must be made as to whether they continue to grow, becoming NTs, or remain at their actual length as filopodia. The signal and mechanism responsible for this decision are not known. Based on these considerations, the formation of membrane nanotubes may be divided into two steps. First, the membrane structures begin to shape up due to the binding of a membrane-sculpturing protein to the lipid bilayer, and short filopodia or filopodium-like structures appear. Next, a cellular decision is made regarding their continued growth to NTs, likely based on the function they should fulfil at the given time and location, and a few of the filopodium-like structures grow further and eventually develop into membrane nanotubes. 

### 3.4. The Role of the IRSp53

Our data indicate that the IRSp53 plays an important role in both of these steps while its I-BAR domain is mostly functional during the first step. We observed that although the number of filopodia increased substantially in both cell lines once I-BAR was overexpressed, the change in the number of membrane nanotubes was not very well-pronounced, especially in A20 cells. This observation indicates that I-BAR is part of the set of proteins initiating the formation of membrane protrusions, rather than continuing to grow from filopodium-like structures to NTs. It is known that I-BAR alone is a strong inducer of membrane shaping and filopodium formation; in addition, by creating stable oligomers, it generates non-dynamic cell protrusions and is evenly distributed at their membrane [38]. During membrane curvature generation, I-BAR forms a continuous coating or layer beneath the membrane by electrostatically interacting with a basic constituent of the plasma membrane (PI(4,5)P2 head carries a large negative charge by which it can bind to the highly positively charged regions of I-BAR). The other side of I-BAR also has an electrostatic interaction with F-actin and even possesses a minimal actin-bundling effect under physiological conditions [6,32,39,40]. The I-BAR itself is not present in the actin bundles; rather, it is localised at the boundary of the actin bundle and the cell membrane [7]. It was revealed by superresolution microscopy that—at least in filopodia—I-BAR situates at the central axis of the protrusions, overlapping with actin filaments, while IRSp53 localises at the sides of the extensions, presumably showing indirect interactions with actin [38]. Here, we reported that the full-length protein IRSp53 that includes the I-BAR domain in its N terminal region is a constituent of the NTs independently of the studied cell type, where it may colocalise with actin filaments. IRSp53 does not simply localise to the tubes but by binding to the membrane, as well as to actin [6,41]; this may promote the build-up of the NTs originated from thin, finger-like actin-driven membrane projections by influencing actin assembly, which can be further supported by the previously mentioned partner of I-BAR, the PI(4,5)P2, due to its ability to promote actin polymerisation by activating formin, Arp2/3 or ezrin, among others [39,42,43]. In addition, biochemical assays proved that, similarly to I-BAR, the full-length IRSp53 itself exhibits a direct binding ability to actin and facilitates the formation of actin bundles [44]. It was reported that in a substoichiometric concentration range, compared to G-actin, IRSp53 slows down barbed end growth [41]; however, superstoichiometric conditions were not measured. The capping activity of IRSp53 requires the presence of the I-BAR and CRIB domains, together with the 30-amino-acid-long sequence prior to the SH3 domain, and it was found that I-BAR alone has no effect on the assembly of actin [41]. It is suggested that IRSp53 binds slowly to the barbed end of actin filaments, assuming that either this binding requires a conformational change in the IRSp53, or the binding induces its structural alterations, leading to a decrease in the rate of the elongation of the filament. On the other hand, IRSp53 has a low affinity to barbed ends, allowing for the growth of the filament following the dissociation of IRSp53 [41]. Here, we showed that the overexpression of IRSp53 led to the generation of more elongated protrusions compared to transfection with the I-BAR domain, suggesting that other domains of IRSp53 may also participate in the growth of membrane projections. It is likely that the SH3 domain of IRSp53 also plays a key role in the formation and elongation of filopodia and NTs through the interaction with actin modulator proteins such as dynamin, Eps8, mDia1, Mena, and N-WASP, which are mainly involved in actin bundling, nucleation or remodelling and were reported to support the formation and regulation of protrusions [2,10,15,45,46,47,48]. Furthermore, IRSp53 cooperates with the actin-bundling protein fascin which was found to promote the elongation of filopodia [45]. Apparently, actin dynamics are precisely regulated in the cells, enabling the induction, elongation and retraction of cell extensions depending on specific cellular conditions. 

### 3.5. Cell Line Differences

We also observed that the change in the number of NTs was different for the two examined cell types. While we observed some increases in the number of NTs of COS-7 cells, the overexpression of proteins in A20 cells had no effect on the NT formation frequency (Figure 6A and Figure 8A). These observations suggest that the typical NT-forming mechanism of a cell may also be an important factor in the growth frequency of NTs. COS-7 cells primarily grow their NTs using the method referred to as ‘making contact’, when a filopodium-like membrane projection initiative elongates to a thin, bridge-like structure by means of actin polymerisation. On the other hand, A20 cells are characterised by the ‘keeping contact’ method of NT initialisation, which presupposes close contact between the cells and their subsequent separation from one another in opposite directions (Appendix A) [17,49]. Our observation may be supported by in silico data that revealed that active forces arising from the actin cytoskeleton and coupled to the intrinsic curvature help with the formation of protrusions [3,50]. Furthermore, it is also likely that I-BAR shows a high affinity to a membrane component that cannot be found in large quantities in A20 cells, and presumably this binding may also play a role in NT formation. Additionally, the difference in the expression levels of I-BAR/IRSp53 may influence the ability of a cell to form NTs. To understand the tendency of the number of cell extensions, it is important to note that the full-length IRSp53 protein consists of several domains, which are able to bind to different actin-binding proteins, meaning that this undergoes a much more complex regulation than the I-BAR domain. The IRSp53 is competitively inhibited by these actin-affector proteins [2,47,51], resulting in a smaller number of filopodium and less NT growth compared to the I-BAR domain.

### 3.6. Model

In light of these results, a simple model can be established to explain the formation of ‘molecular relay stations’ along the membrane nanotubes. The model is summarised in Figure 11 and suggests that the I-BAR domain of the IRSp53 is mainly responsible for the creation of the membrane curvature at the beginning of the protrusion formation, and then also plays a role in the maintenance of the membrane shape. In silico studies suggest that curved membrane proteins with a convex shape (such as I-BAR) tend not only to aggregate but also to recruit the cytoskeleton to promote the formation and/or elongation of cell protrusions, and consequently help in the fulfilment of cellular function by shaping the cell. The coupling of the deforming proteins with the active cytoskeletal elements provides the necessary force to deform the membrane [3]. The IRSp53 is involved in the elongation of the membrane extensions. The IRSp53 is localised along the membrane protrusions and nucleates de novo actin polymerisation. The actin nuclei formed by the IRSp53 serve as new locations for the elongation of actin filaments, and the newly polymerising actin filaments join and strengthen the growing actin bundles. These molecular relay stations can provide the mechanisms by which the number of actin filaments is kept constant regardless of the end in the growth in some of the participating polymers. At the same time, by being anchored to the membrane, they can also serve as mechanical supporting points for the growth and force generation of the actin filament bundles along the membrane nanotubes. Our model is supported by the observation that, in the growing protrusions, IRSp53 can accumulate in dense foci, which is likely to aid in the early phase of filopodia [41,52,53] and NT [45] formation. Furthermore, simulations predicted that I-BAR domains also tend to form aggregates in the protrusions to stabilise them [3,54]. The accumulation of I-BAR/IRSp53 in discrete puncta in cell protrusions suggests that the ratio of actin: I-BAR or actin: IRSp53 locally shifted towards the increased I-BAR or IRSp53 concentration (compared to actin), promoting the elongation of the membrane extensions. All of this is in good agreement with previous studies discussing how the formation of IRSp53 clusters contributes to the further recruitment of IRSp53 and promotes actin assembly in NTs; furthermore, in the absence of IRSp53 enrichment, NTs lack F-actin [45]. 

We assume that our model is of particular importance as the presence of actin is essential for the formation and maintenance of filopodia generated by IRSp53 [38]. Given the critical biomedical significance of the actin-rich protrusions investigated in our study, the precise regulation and fine-tuning of actin dynamics that impact the life cycle of these cellular extensions (via mechanisms such as promoting actin nucleus formation, actin bundling or capping of actin filaments) hold vital physiological importance.

## 4. Materials and Methods

### 4.1. Protein Purification and Labelling

I-BAR domain of IRSp53 was expressed in bacterial expression system; IRSp53 was procured from MyBioSource (San Diego, CA, USA). For bacterial protein expression, cDNAs of the IRSp53 I-BAR domain were inserted into pGEX-4T2 vector (Amersham Biosciences, Piscataway, NJ, USA), then expressed as glutathione *S*-transferase (GST) fusion proteins in *Escherichia coli* BL21 strain (Novagen, Darmstadt, Germany). Bacteria were grown at 37 °C in Luria Broth powder microbial growth medium (Sigma-Aldrich, St. Louis, MO, USA) following the usual transformation procedure. Protein expression was induced by the addition of 1 mM isopropyl β-d-1-thiogalactopyranoside (IPTG, Sigma-Aldrich, St. Louis, MO, USA) at OD_600nm_ ~0.4–0.6 (Jasco V660) prior to overnight expression at 25 °C, after which the bacterial extracts were collected by centrifugation (6000× *g*, 4 °C, 10 min) and stored at −80 °C until further use in aliquots of 6 g. Prior to purification, the bacterial pellet was washed in buffer L [lysis buffer (50 mM tris(hydroxymethyl)aminomethane/Tris-Base, Sigma-Aldrich, St. Louis, MO, USA/, pH 7.5, 300 mM NaCl/Molar Chemicals, Hungary/, 3 mM dithiothreitol/DTT, Duchefa Biochemie, Haarlem, Netherlands/, 1 mM ethylenediaminetetraacetic acid /EDTA, Sigma-Aldrich, St. Louis, MO, USA/and 2.5% glycerol/Molar Chemicals, Hungary/)], supplemented with Protease Inhibitor, 8 mg lysozyme and 0.5 mM phenylmethylsulfonyl fluoride (PMSF, all from Sigma-Aldrich, St. Louis, MO, USA), then was homogenised on ice. For protein purification, homogenised cells were lysed by sonication (5 × 1 min, with 80% amplitude, on ice), then stirred for 1 h at 4 °C in the presence of 0.1 mg/mL DNase (PanReac-Applichem, Barcelona, Spain). The cell lysate was ultracentrifuged (100,000× *g*, 4 °C, 30 min), and the supernatant was slowly loaded onto glutathione (GSH) resin (5 mL, Amersham Biosciences, Piscataway, NJ, USA) in a column. The column was washed with lysis-, then I-BAR buffers (4 mM Tris-Base /Sigma-Aldrich, St. Louis, MO, USA/ pH 6.8, 0.1 mM CaCl_2_ /Serva, Heidelberg, Germany/, 0.5 mM 2-mercaptoethanol /BME/, 0.005% (*w*/*v*) sodium azide /NaN_3_, both from Sigma, St. Louis, MO, USA/ 0.1% (*w*/*v*) sucrose /Sigma, St. Louis, MO, USA/, 2.5% (*v*/*v*) glycerol /Molar Chemicals, Hungary/ and 150mM KCl /Scharlab, Barcelona, Spain/). To cleave the GST fusion tag from the protein, the sample was incubated overnight at 4 °C on the column with 1.4 mg/mL thrombin (Sigma-Aldrich, St. Louis, MO, USA), then washed with I-BAR buffer. Purification of the protein was then conducted via suspension of protein sample with benzamidine beads (GE Healthcare, Chicago, IL, USA) for 1 h at 4 °C, followed by several wash steps with I-BAR buffer. Finally, the eluted protein was concentrated (Corning Spin-X UF 5KDa concentrator, Sigma-Aldrich, St. Louis, MO, USA), then flash-frozen in liquid nitrogen and stored at −80 °C until use. The protein concentration was measured spectrophotometrically using the extinction coefficients at 280 nm and molecular weight derived from the amino acid sequence (ExPASy ProtParam tool http://web.expasy.org/protparam/ (accessed on 27 July 2023)) [55]. Typically, 5–6 g of bacterial pellet yielded 3.5–4.0 mg/mL protein. 

Actin was isolated from rabbit skeletal muscle according to the protocol described earlier by Spudich and Watt [56] and was further purified by gel filtration on Superdex 200 column (GE Healthcare, Chicago, IL, USA), and stored in G buffer (4 mM Tris-HCl /Sigma-Aldrich, St. Louis, MO, USA/, pH 7.8, 0.1 mM CaCl_2_ /Serva, Heidelberg, Germany/, 0.2 mM ATP /Sigma-Aldrich, St. Louis, MO, USA/, 0.005% (*w*/*v*) NaN_3_ /Sigma-Aldrich, St. Louis, MO, USA/ and 0.5 mM BME/Sigma-Aldrich, St. Louis, MO, USA/) [57]. For TIRFM measurements, G-actin was labelled at Lys^328^ by Alexa Fluor^®^ 488 carboxylic acid succinimidyl ester (Alexa488NHS, Thermo Fisher Scientific, Waltham, MA, USA); for the measurement of the kinetics of actin assembly, G-actin was labelled at Cys374 by N-(1-pyrenyl)iodiacetamide (Thermo Fisher Scientific, Waltham, MA, USA) [58].

### 4.2. Cells, Culture, Cell Transfection

COS-7 (ATCC Cat# CRL-1651, RRID: CVCL_0224) *Cercopithecus aethiops* fibroblast-like kidney cells were cultured in Dulbecco’s modified Eagle’s medium (DMEM, Sigma Aldrich, St. Louis, MO, USA) supplemented with 10% fetal bovine serum (FBS) at 37 C°, in 5% CO_2_ thermostat on special borosilicate chamber microplate wells (MatTek, Ashland, MA, USA) at closely physiological conditions (37 ± 0.1 °C; CO_2_: 5%). A typical density of 25,000 cells/cm^2^ was used in experiments. A20 mature B cells (ATCC Cat# TIB-208, RRID: CVCL_1940) were cultured at 37 °C in 5% CO_2_ incubator in RPMI-1640 medium (Pan-Biotech, Aidenbach, Germany), supplemented with 2 mM L-glutamine (Lonza, Basel, Switzerland), 1mM Na-pyruvate (Lonza, Basel, Switzerland), antibiotics and 10% FBS (Sigma Aldrich, St. Louis, MO, USA). Typically, a cell density of 3 × 10^5^ cells/cm^2^ was used for further experimentations.

For live cell experiments, cells were transfected either with mCherry-IRSp53 or mCherry-I-BAR constructs (for plasmid origin, see https://etheses.bham.ac.uk//id/eprint/860/ (accessed on 27 July 2023), [59]) using Lipofectamine 3000 (Thermo Fisher Scientific, Waltham, MA USA) in full accordance with the manufacturer’s recommended guidance (COS-7 cells) or were electroporated using an Amaxa Nucleofector IIb device (program L-013 /A20/ was used), in full accordance with the manufacturer’s recommended guidance. GFP-LifeAct plasmid (Ibidi, Martinsried, Germany) was used and transfected with the same procedure in control experiments. In each transfection, 1 (IRSp53 and I-BAR)—2 (LifeAct) µg of plasmids were used. Cells were visualised under live-cell conditions 16 h following transfection.

### 4.3. Immunocytochemistry and Cell Staining

Cells were fixed with 4% paraformaldehyde (PFA, Alfa Aesar, Haverhill, MA, USA) for 10 min at room temperature (RT), then permeabilised with 0.1% Triton X-100 + 5% bovine serum albumin (BSA, both from Sigma-Aldrich, St. Louis, MO, USA) for 20 min at RT and incubated with anti-IRSp53 antibody (1:500; Santa Cruz Biotechnology, Dallas, TX, USA, catalogue number: sc-134810) for 1 h at RT, then washed and labelled with secondary antibody (1:1000; goat anti-rabbit Alexa Fluor 488; Thermo Fisher Scientific, Waltham, MA, USA, Cat# A-11008, RRID: AB_143165) and sometimes with Alexa 488- or 561-phalloidin (Life Technologies, Carlsbad, CA, USA). Hoechst (1:1000; Biotium, Fremont, CA, USA) was used to label cell nuclei. Samples were mounted with VECTASHIELD Antifade Mounting Medium (Vector Laboratories, Burlingame, CA, USA), then stored at 4 °C in the dark until visualisation. The functionality of NTs was verified by Alexa488-Cholera toxin B (CTX-B) labelling as described previously [16].

### 4.4. Microscopic Imaging

The effect of transfections on cells and protrusive structures was visualised using a Zeiss LSM 710 Confocal Laser Scanning Microscope (CLSM, d_lat_: ~250 nm) at 63× magnification (oil immersion objective; N.A.: 1.4). Records were taken and analysed with the Zen black 2.1 SP3/Zen blue 2.3 software and further analysed using Image J (FIJI; Wayne Rashband, NIH, Washington, USA (RRID: SCR_002285), https://fiji.sc/ (accessed on 27 July 2023)) and, occasionally, Imaris 8.2 (Bitplane, Zürich, Switzerland (RRID: SCR_007370)) software programs. To quantify the relation between the position of IRSp53 with F-actin in NTs, colocalisation was performed within a defined ROI along the entire length of the NT. The positional relation of IRSp53 and actin under the cell membrane was measured within an ellipsoidal ROI covering nearly half the circumference of the cell membrane, and colocalisation of the proteins along pronounced stress fibres was quantified within the ROI of elongated ellipsis around the actin bundles. Colocalisation coefficients M_1_ and M_2_ were calculated by Zen black 2.1 SP3 software (RRID: SCR_018163), based on the formulas described by Manders et al. [60] as follows:(1)$$M_{1}\left(\%\right)=\frac{number\;of\;pixels\;in\;channel\;green\;\left(i.e.\;\;IRSp53\right)\;colocalised}{number\;of\;pixels\;in\;channel\;green\;\left(i.e.\;\;IRSp53\right)\;in\;total}*100$$
and
(2)$$M_{2}\left(\%\right)=\frac{number\;of\;pixels\;in\;channel\;red\;\left(i.e.\;\;F-actin\right)\;colocalised}{number\;of\;pixels\;in\;channel\;red\;\left(i.e.\;\;F-actin\right)\;in\;total}*100.$$

### 4.5. Statistical Analysis

The relative frequency of NTs/filopodia was calculated from a minimum of three independent experiments from at least ~800 cells/sample and given as the ratio of cells forming at least one NT/filopodia per all cells visible in the particular field (mean ± SD). Morphological parameters of the cell projections were manually tracked. The length of NTs was defined as the distance between the growth cones of the NTs, the thickness (i.e., diameter) of NTs was measured at three points in each tube (directly behind the growth cones and in the middle of the tube); next, the thickness was calculated as the arithmetic mean of the measured data [61]. NT branching was defined as a division or subdivision from the main axis of a NT. The length of NT branching was given as the distance between the branching point in the NT and either the joining point in the neighbouring cell or the tip of the branching. The length of filopodia was determined as the distance between the leading edge of the cell and the tip of the filopodium. The image analysis and the statistical calculations, including significance tests (using Student *t*-test or Mann–Whitney test), were carried out using ImageJ/FIJI (Wayne Rashband, NIH, Washington, DC, USA), Origin 2020 (OriginLab Corporation Northampton, MA, USA (RRID: SCR_014212)) or IBM SPSS Statistics (version 26, IBM SPSS Statistics, Armonk, NY, USA (RRID: SCR_019096)) statistical programs, respectively. If the results of Kolmogorov–Smirnov normality test disproved normal distribution, post hoc Kruskal–Wallis tests (or, for normal distribution, ANOVA tests) were performed to compare each treatment to the control. Mann–Whitney U test (or, in normal distribution, Student *t*-test) was used to avoid type II errors and to confirm the significance suggested by Kruskal–Wallis analysis. The level of significance was set at *p* < 0.05.

### 4.6. Total Internal Reflection Fluorescence Microscopy

TIRFM was used to follow the effect of either the I-BAR domain or the IRSp53 protein on the assembly of individual actin filaments as described previously [62]. Glass flow cells were incubated with 100 µL of N-ethylmaleimide myosin for 1 min, then washed with 200 µL of myosin buffer (4 mM Tris-HCl /Sigma-Aldrich, St. Louis, MO, USA/pH 7.8, 1 mM DTT /Duchefa Biochemie, Haarlem, Netherlands/, 0.2 mM ATP /Sigma-Aldrich, St. Louis, MO, USA/, 0.1 mM CaCl_2_ /Serva, Heidelberg, Germany/, 500 mM KCl /Sigma-Aldrich, St. Louis, MO, USA/, 1 mM MgCl_2_ /Sigma-Aldrich, St. Louis, MO, USA/, 0.2 mM ethylene glycol-bis(β-aminoethyl ether)-N,N,N′,N′-tetraacetic acid /EGTA, Sigma-Aldrich, St. Louis, MO, USA/) and 200 µL of 0.1% (*w*/*v*) BSA (Sigma-Aldrich, St. Louis, MO, USA). 200 µL of TIRFM buffer (0.5% (*w*/*v*) methylcellulose /Sigma-Aldrich, St. Louis, MO, USA/, 0.5% (*w*/*v*) BSA /Sigma-Aldrich, St. Louis, MO, USA/, 50 mM 1,4-diazabicyclo-[2,2,2]octane /DABCO, Sigma-Aldrich, St. Louis, MO, USA/, 100 mM DTT /Duchefa Biochemie, Haarlem, Netherlands/ in buffer F* (4 mM Tris-HCl, pH 7.8, 1 mM DTT, 0.2 mM ATP, 0.1 mM CaCl_2_, 50 mM KCl, 1 mM MgCl_2_ and 0.2 mM EGTA) was used in the flow cells prior to the addition of the protein mixture (for exact protein composition and concentration, see Figure legends). The assembly of free G-actin in the absence and presence of I-BAR/IRSp53 was visualised in the flow cell by mixing I-BAR/IRSp53 and G-actin (0.5 µM, containing 10% Alexa488NHS-G-actin, Thermo Fisher Scientific, Waltham, MA, USA) in TIRFM buffer. The images were captured every 10 s with a laser-based (491 nm) TIRFM module (APON TIRF 60×, N.A.: 1.45 oil immersion objective) set up on an Olympus IX81 microscope using a CCD camera (Hamamatsu /Sizuoka, Japan/). TIRFM images were analysed using FIJI software (https://fiji.sc/ (accessed on 27 July 2023)) [63], implementing the following procedure. The same region of interest (724 × 724 pixels) was used in all images, following background subtraction (rolling ball radius was set to 50 pixels, sliding paraboloid method was applied), 1% threshold was adjusted. A minimum size of five pixels was used during particle analysis to exclude noise. Analysed and raw images were compared, and manual correction was carried out to avoid discrepancies. 

### 4.7. Pyrenyl Polymerisation Assay

Mg^2+^-ATP-G-actin polymerisation kinetics were measured in pyrenyl polymerisation assay. Polymerisation was launched in the presence of 2 mM MgCl_2_ and 100 mM KCl (both from Sigma-Aldrich, St. Louis, MO, USA) with 2.5 µM G-actin containing 5% pyrenyl-labelled actin in the absence or presence of different concentrations of I-BAR domain. The assembly of actin was followed up by the change in the emission of the pyrenyl fluorescence in a Safas Xenius FLX spectrofluorimeter (using 3 nm slit in the excitation and 5 nm slit in the emission side, with $\lambda_{ex}=365\;nm$, $\lambda_{em}=404\;nm$ and 720 V). Due to the low concentration of the recombinant IRSp53, we could not perform polymerisation assays with the full-length protein.

### 4.8. General Experimental Conditions

Samples at each concentration were prepared separately for experiments. All TIRFM and actin polymerisation measurements were performed at 20 °C. The sum of the volume of the proteins and the volume of their storing buffer was constant in the samples and represented a maximum of 50% of the total volume of the sample. The concentrations given in the text are final concentrations. In all experiments, Mg^2+^-ATP-actin was used. The actin monomer bound Ca^2+^ was replaced by Mg^2+^ by the addition of 200 µM EGTA and 50 µM MgCl_2_ and the samples were incubated for 5 min at RT. The data presented were derived from at least three independent experiments. Confocal images were captured under live-cell imaging conditions (37 °C; 5% CO_2_ atmosphere), and optical slicing was used to clearly distinguish the NTs (freely hovering in the culture medium) from the filopodia (attached to the bottom of the culture dish) in a quantitative analysis (Appendix A), furthermore, we made time-lapse recordings in ambiguous cases (Appendix A).

## 5. Conclusions

The present data illustrate that the IRSp53 protein, together with other regulatory protein complexes, can form an essential network that can regulate dynamic actin polymerisation by deforming membrane, forming a nucleation site and mechanically stabilising actin bundles. These molecular events appear to be essential for the inception of various nanotubular networks that connect cells. Since these ancient NT networks recently are the focus of attention concerning understanding of the cell-to-cell communication pathways during the evolution and in health and diseases [64]. We believe these new results will prove helpful in designing new, tunnelling-NT-based therapeutic modalities.

## Figures and Tables

**Figure 1 ijms-24-13112-f001:**
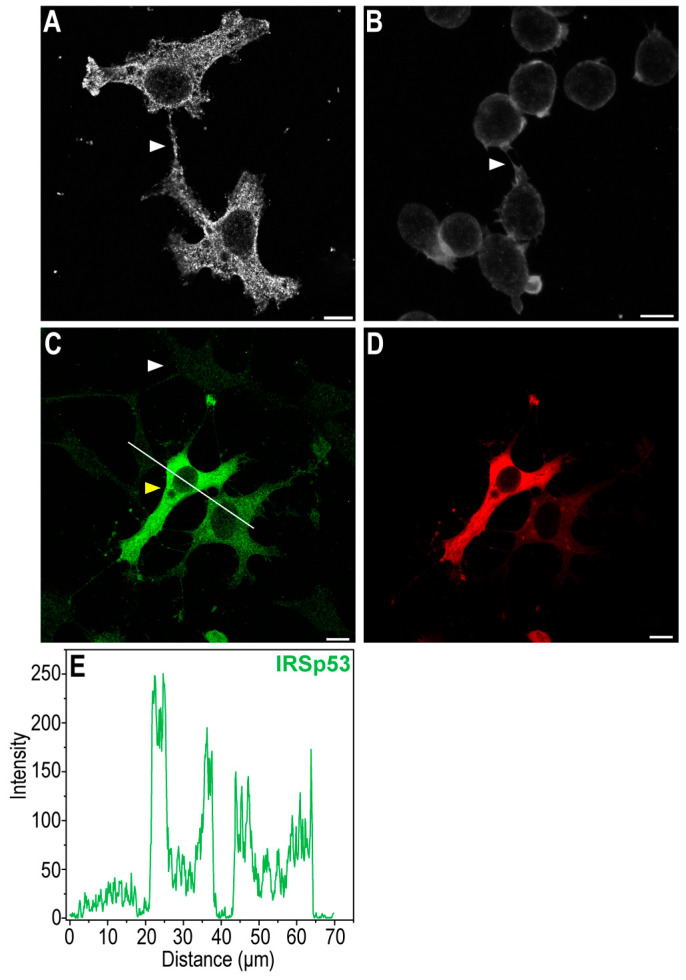
IRSp53 protein is endogenously expressed both in COS-7 kidney and in A20 B cells, even in their NTs. (**A**) CLSM image shows the focal plane of the NT in COS-7 cells (NT is depicted by an arrowhead). (**B**) Representative CLSM image of the focal plane of the NT in A20 cells (NT is depicted by an arrowhead). (**C**) The IRSp53 pattern of non-transfected (i.e., endogenous IRSp53 expression, white arrowhead) and transfected (i.e., overexpressed IRSp53, yellow arrowhead) COS-7 cells, image was captured using CLSM with the transfected cells in focus, the cells were first transfected with mCherry-IRSp53 (which is shown in (**D**)), then fixed and labelled for the presence of IRSp53 in an immunofluorescence assay. Remarkably, the transfection efficiency is far from 100%. (**E**) The line scan analysis shows the difference in the intensities of IRSp53 in non-transfected and transfected cells; the analysis was performed along the white line in (**C**). CLSM images were captured at 63× magnification (N.A.: 1.4). In immunofluorescence assays (**A**–**C**) cells were fixed and then labelled for the presence of IRSp53. Scale bar: 10 µm.

**Figure 2 ijms-24-13112-f002:**
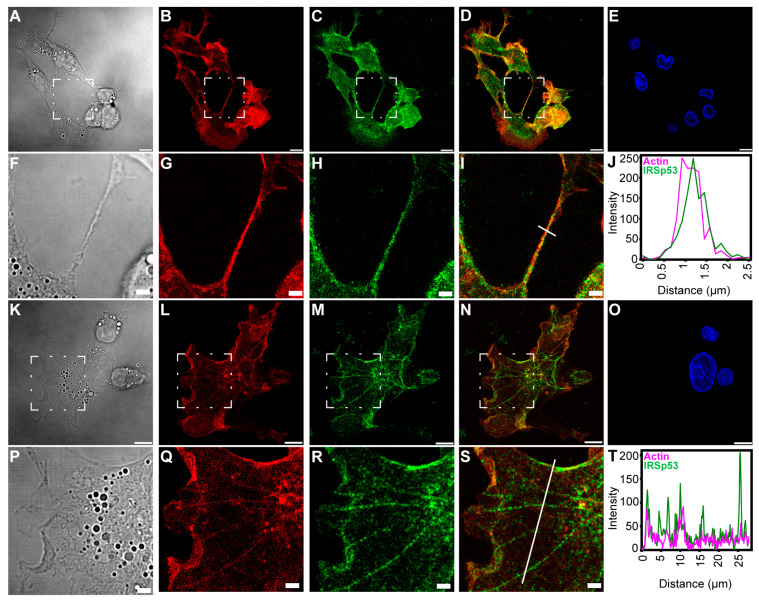
Colocalisation of actin and IRSp53 in COS-7 cells. (**A**–**J**) show the positional relation between IRSp53 and actin in a well-resolved NT of COS-7 cells; (**A**,**F**) DIC images, where (**F**) is the zooming of the boxed region of (**A**); (**B**,**G**) actin, (**G**) is zoomed from the boxed area of (**B**); (**C**,**H**) IRSp53, (**H**) is the zooming of the boxed region in (**C**); (**D**,**I**) merged images of actin and IRSp53, where (**I**) was zoomed from the boxed area of (**D**); (**E**) nucleus; (**J**) a line scan analysis indicates the intensity distribution along the NT, from which the approximate thickness of the NT can be evaluated; the analysed region is shown by the white line in (**I**). (**K**–**T**) show the positional colocalisation of IRSp53 and actin under the membrane and along the actin-based stress fibres in COS-7 cells; (**K**,**P**) DIC images, where (**P**) is the zooming of the boxed region of (**K**); (**L**,**Q**) actin, (**Q**) is zoomed from the boxed area of (**L**); (**M**,**R**) IRSp53, (**R**) is the zooming of the boxed region in (**M**); (**N**,**S**) merged images of actin and IRSp53, where (**S**) was zoomed from the boxed area of (**N**); (**O**) nucleus; (**T**) the intensity profile of the examined proteins shows a relation between IRSp53 and actin; the analysed region is shown by the white line in (**S**). Actin was labelled with Alexa 561-phalloidin, and immunocytochemistry was used to label endogenous IRSp53. Microscopic images were captured using CLSM at 63× magnification (N.A.: 1.4). Scale bars: 10 µm (**A**–**E**,**K**–**O**) and 3 µm (**F**–**I**,**P**–**S**), respectively.

**Figure 3 ijms-24-13112-f003:**
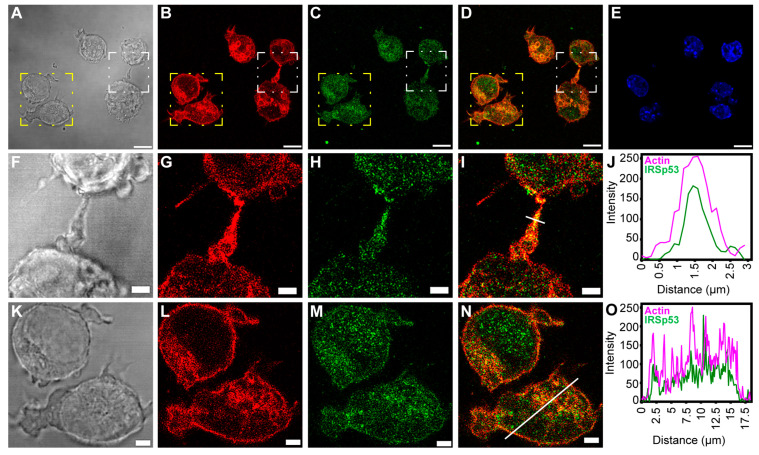
Colocalisation of actin and IRSp53 in A20 cells. (**A**,**J**) show the positional relation of IRSp53 and actin in a NT of A20 cells; (**A**,**F**) DIC images, where (**F**) is the zooming of the white boxed region of (**A**); (**B**,**G**) actin, (**G**) is zoomed from the white boxed area of (**B**); (**C**,**H**) IRSp53, (**H**) is the zooming of the white boxed region in (**C**); (**D**,**I**) merged images of actin and IRSp53, where (**I**) was zoomed from the white boxed area of (**D**); (**E**) nucleus; (**J**) the intensity profile of the examined proteins shows a comparable distribution of IRSp53 and actin; the estimated thickness of the NT can be determined from axis *x* (the analysed region is shown in (**I**) by the white line). (**K**,**O**) show the presumed colocalisation of IRSp53 and actin in A20 cells; (**K**) DIC, zooming of the yellow boxed region of (**A**); (**L**) actin, zoomed from the yellow boxed area of (**B**); (**M**) IRSp53, magnification of the yellow boxed region of (**C**); (**N**) merged image of actin and IRSp53, zooming of the yellow boxed region in image (**D**); (**O**) the line scan analysis shows the intensity profile of IRSp53 and actin, suggesting a similar occurrence pattern with a highly increased intensity under the cell membrane (the analysed region is shown in (**N**) by the white line). Actin was labelled with Alexa 561-phalloidin, and immunocytochemistry was performed to label endogenous IRSp5. Microscopy was performed on CLSM, at 63× magnification (N.A.: 1.4). Scale bars: 10 µm (**A**–**E**) and 3 µm (**F**–**I**,**K**–**N**), respectively.

**Figure 4 ijms-24-13112-f004:**
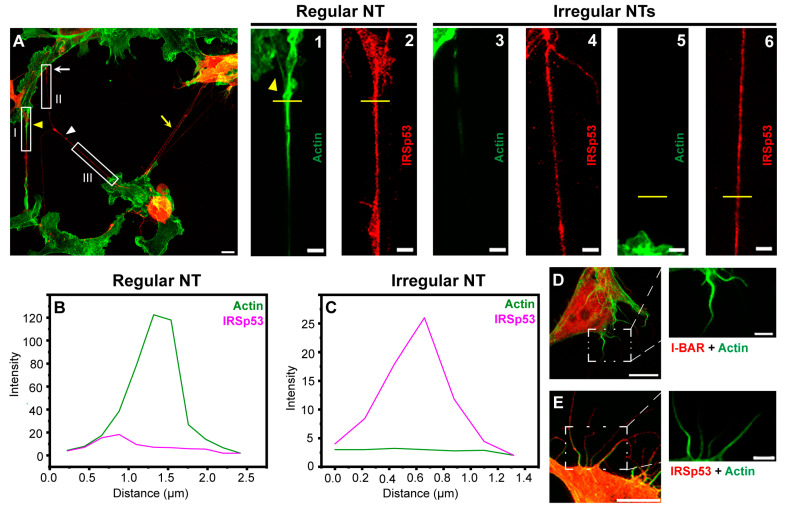
Actin pattern of NTs and filopodia of transfected COS-7 cells. (**A**) Representative CLSM recording of IRSp53 transfected and actin labelled cells displays that the overexpression of IRSp53 resulted in the formation of not only NTs with usual morphology (yellow arrowhead), but an entangled nanotubular network (yellow arrow), as well as long, thin, NT-like projections with an irregular shape (white arrowhead), without observable growth cone (white arrow). Only NTs displaying the signal of mCherry-IRSp53 were analysed; scale bar: 10 µm. (**1**,**2**) The magnified views of boxed region I show that NT characteristic for COS-7 cells has pronounced growth cone (yellow arrowhead) and contains actin in the entire length; scale bars: 3 µm. (**3**,**4**) The magnified views of boxed region II show that the pattern of F-actin of the irregular membrane projection is clearly different from that of regular NT; these tangled protrusions are enriched with less, sometimes patchy F-actin; scale bars: 3 µm. (**5**,**6**) are zoomings of boxed region III, scale bars: 3 µm. (**B**,**C**) While the intensity of actin in the NT-like structures of COS-7 cells showed high diversity, the amount of IRSp53 proved to be more uniform (see intensity values on y axes); the analysed regions are shown by the yellow lines in panels (**1**,**2**) for regular NT, and (**5**,**6**) for irregular NT. (**D**,**E**) Filopodia of either I-BAR or IRSp53 transfected cells contain pronounced F-actin bundles, scale bars: 10 µm (magnifications of the boxed areas show the actin pattern alone, scale bars: 3 µm). Microscopic images were captured using CLSM at 63× magnification (N.A.: 1.4).

**Figure 5 ijms-24-13112-f005:**
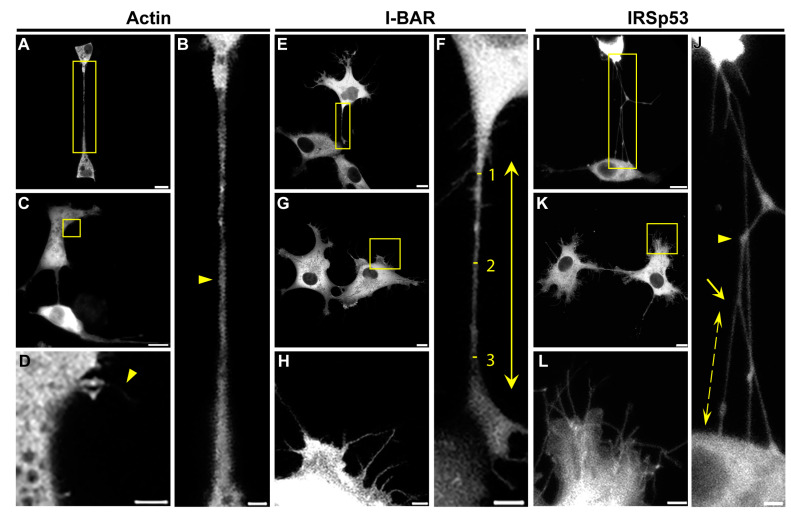
Effects of actin, I-BAR and IRSp53 overexpression on COS-7 cells. (**A**,**C**) LifeAct-GFP was used as a control in the experiments. (**B**) The magnification of the yellow boxed region of image (**A**) indicates that NTs connecting LifeAct-GFP expressed cells (yellow arrowhead) did not show any branching or protein accumulations inside their tubes. (**D**) The magnification of the yellow boxed region of image (**C**) displays actin transfected cells formed of only a few filopodia (yellow arrowhead). (**E**,**G**) The effect of the overexpression of mCherry-I-BAR on COS-7 cells. (**F**) The zooming of the boxed region of image (**E**) shows that NTs of I-BAR transfected cells contained branches, as well as protein accumulations. The length of NTs was determined as the distance between the growth cones of the NTs (yellow arrow); furthermore, the thickness of NTs was measured at three points in a tube: directly behind the growth cones and in the middle of the NT (yellow lines and numbers), and the diameter was calculated as the average of the measured values. (**H**) The zooming of the yellow boxed region of image (**G**) shows a higher frequency of filopodia formation. (**I**,**K**) Representative CLSM images of mCherry-IRSp53 expressed COS-7 cells. (**J**) The magnified view of the boxed region of image (**I**) illustrates that IRSp53 transfected cells grew membrane nanotubes with branches (yellow arrow), and protein accumulations, where tubes often changed direction (yellow arrowhead). The length of NT branching was measured between the branching point in the NT and the joining point in the neighbouring cell (yellow scattered arrow). (**L**) The magnified view of the boxed region of image (**K**) displays that COS-7 cells formed significantly more filopodia after IRSp53 transfection. Microscopic images were captured using CLSM at 63× magnification (N.A.: 1.4). Scale bars: 10 µm (**A**,**C**,**E**,**G**,**I**,**K**) and 3 µm (**B**,**D**,**F**,**H**,**J**,**L**), respectively.

**Figure 6 ijms-24-13112-f006:**
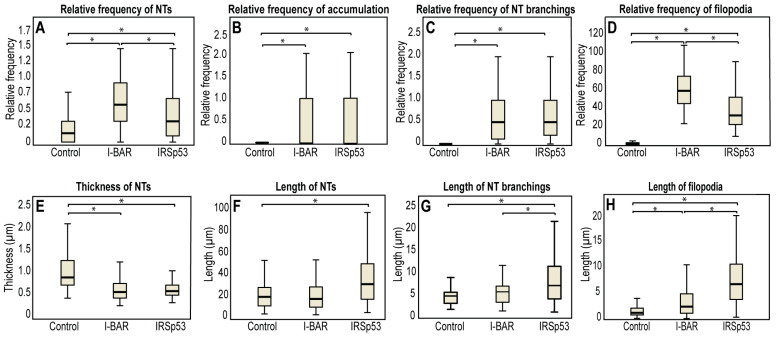
Morphological changes in membrane nanotubes and filopodia of COS-7 cells due to I-BAR and IRSp53 overexpression. (**A**) The frequency of NT formation in control and I-BAR/IRSp53 overexpressed cells. (**B**) The frequency of accumulation puncta significantly increased in I-BAR and IRSp53 transfected cells. (**C**) The change in the frequency of NT branchings following the overexpression of I-BAR and IRSp53. (**D**) The frequency of the formation of filopodia in control and I-BAR/IRSp53 overexpressed cells. (**E**) The change in the thickness of NTs after the overexpression of I-BAR and IRSp53. (**F**) The alteration in NT length following I-BAR and IRSp53 transfection. (**G**) The change in the length of NT branchings following the overexpression of I-BAR and IRSp53. (**H**) The length of filopodia in control and I-BAR/IRSp53 transfected cells. Actin transfection was used as control in the study. Statistical analysis was carried out for three independent examinations, in which at least 800 cells were visualized in each experiment. The frequency of NT/filopodia formation was calculated as follows: $\frac{\mathrm{number}\;\mathrm{of}\;\mathrm{nanotubes}/\mathrm{filopodia}\;\mathrm{in}\;\mathrm{the}\;\mathrm{FOV}}{\mathrm{all}\;\mathrm{cells}\;\mathrm{visible}\;\mathrm{in}\;\mathrm{the}\;\mathrm{particular}\;\mathrm{FOV}}$. The frequency of accumulations or branchings was given as: $\frac{\mathrm{number}\;\mathrm{of}\;\mathrm{accumulations}/\mathrm{branchings}\;\mathrm{in}\;\mathrm{the}\;\mathrm{FOV}}{\mathrm{all}\;\mathrm{NTs}\;\mathrm{in}\;\mathrm{the}\;\mathrm{particular}\;\mathrm{FOV}}$. Results are shown as mean ± SD, *p* < 0.05, indicated with an asterisk.

**Figure 7 ijms-24-13112-f007:**
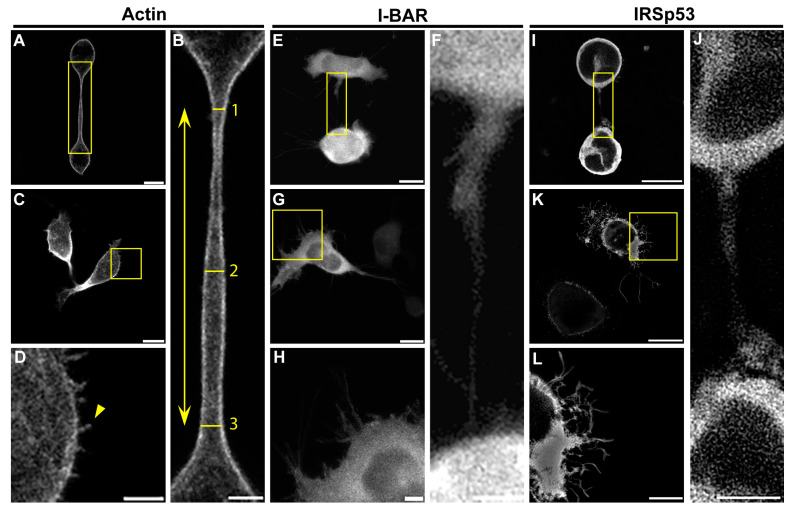
Representative confocal images regarding the effect of I-BAR and IRSp53 overexpression on A20 cells. (**A**,**C**) LifeAct-GFP expressed A20 cells. The NT-forming ability of control cells is relatively low (~10%), and these tubes are typically straight without any branches or breakages. (**B**) The cropping from the yellow boxed region of image (**A**) shows the morphology of an open-ended tunnelling membrane nanotube. The length of NTs was determined as the distance between the growth cones of the NTs (yellow arrow); the thickness of NTs was measured at three points in a tube: directly behind the growth cones and in the middle of the NT (yellow lines and numbers). Additionally, the diameter was calculated as the arithmetic average of the measured values. (**D**) The magnified view of the boxed region of (**C**) indicates A20 cells grow short, spike-like filopodia (yellow arrowhead). (**E**,**G**) show mCherry-I-BAR-overexpressed A20 cells. (**F**) The magnification from image (**E**) boxed region. (**H**) The zooming of the boxed region in (**G**). (**I**,**K**) The effect of the IRSp53 overexpression. (**J**) The magnification of image (**I**) boxed region. (**L**) The magnification of image (**K**) boxed region. Microscopic images were captured using CLSM at 63× magnification (N.A.: 1.4). Scale bars: 10 µm (**A**,**C**,**E**,**G**,**I**,**K**) and 3 µm (**B**,**D**,**F**,**H**,**J**,**L**), respectively.

**Figure 8 ijms-24-13112-f008:**
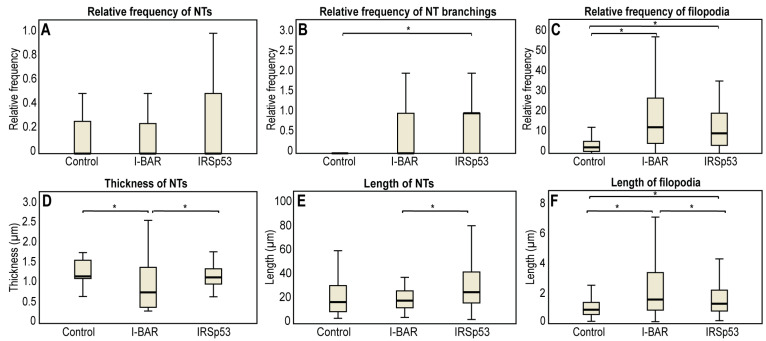
Statistical analysis of the effect of I-BAR and IRSp53 overexpression on the morphological changes of membrane nanotubes and filopodia of A20 cells. While in COS-7 cells, IRSp53 and its I-BAR domain induced similar changes on the examined cell protrusions, in A20 cells these effects were not as obvious. (**A**) The frequency of NT formation in control and I-BAR/IRSp53 overexpressed A20 cells. (**B**) The change in the frequency of NT branchings due to the overexpression of I-BAR and IRSp53. (**C**) The frequency of filopodia formation in control and I-BAR/IRSp53 overexpressed A20 cells. (**D**) The change in the thickness of NTs following the overexpression of I-BAR and IRSp53. (**E**) The alteration in NT length following I-BAR and IRSp53 transfection. (**F**) The length of filopodia in control and I-BAR/IRSp53 transfected A20 cells. Relative frequency of NTs/filopodia was calculated as: $\frac{\mathrm{cells}\;\mathrm{forming}\;\mathrm{at}\;\mathrm{least}\;\mathrm{one}\;\mathrm{nanotube}\;\mathrm{or}\;\mathrm{filopodium}\;}{\mathrm{all}\;\mathrm{cells}\;\mathrm{visible}\;\mathrm{in}\;\mathrm{the}\;\mathrm{particular}\;\mathrm{FOV}}$. Frequency of branchings was given as: $\frac{\mathrm{number}\;\mathrm{of}\;\mathrm{NT}\;\mathrm{branchings}\;\mathrm{in}\;\mathrm{the}\;\mathrm{FOV}}{\mathrm{all}\;\mathrm{NTs}\;\mathrm{in}\;\mathrm{the}\;\mathrm{particular}\;\mathrm{FOV}}$. *p* < 0.05, indicated with an asterisk.

**Figure 9 ijms-24-13112-f009:**
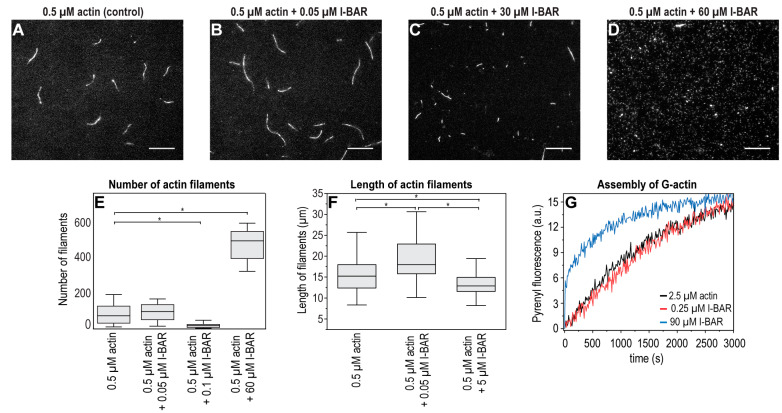
Concentration-dependent effects of I-BAR domain on actin polymerization. TIRFM images were taken every 400 s following the initiation of assembly. Each reaction contained 0.5 μM G-actin (labelled by 10% Alexa488NHS) in TIRFM buffer and various concentrations of the I-BAR domain. Only elongated objects were analysed. (**A**) In the absence of I-BAR (i.e., control), filaments of the same length were formed with minimally shorter filaments in the background. (**B**–**D**) As the stoichiometric ratio of I-BAR increased, the length of actin filaments decreased and the number of minuscule filament initiatives increased significantly; at the concentration of 60 μM of I-BAR (i.e., actin:I-BAR = 1:120), no typical filament was found, and only their initiatives covered the surface. (**E**,**F**) Statistical analysis of TIRF micrographs. (**G**) Representative polymerization kinetics of actin (2.5 µM, containing 5% pyrenyl-labelled actin) in the absence and the presence of a sub- (0.25 µM) and a superstoichiometric (90 µM) concentration of I-BAR domain. We could not measure the effects of a higher concentration of I-BAR on actin polymerisation in our experimental setup. Scale bars: 10 µm, *p* < 0.05, significance compared to the control (actin alone) is indicated with an asterisk.

**Figure 10 ijms-24-13112-f010:**
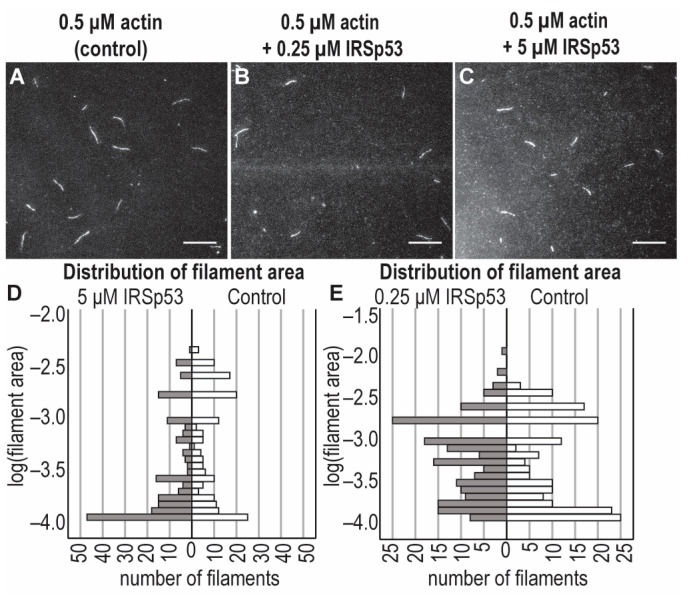
Effects of the full-length IRSp53 protein on actin assembly in TIRFM experiments. TIRFM images were captured at 400 s following the initiation of the actin polymerisation. Polymerisation was performed with 0.5 μM actin (10% Alexa488NHS labelled) in the presence of different concentrations of IRSp53 (0 μM, 0.25 μM and 5 μM). (**A**–**C**) Representative TIRF micrographs of the effects of sub- and superstoichiometric concentrations of IRSp53 on actin filaments. (**D**,**E**) Statistical analysis of the size distribution of the filaments of TIRF micrographs revealed that substoichiometric concentrations of IRSp53 (actin:IRSp53 = 2:1) principally promoted the formation of medium-length filaments, while the presence of IRSp53 in high concentrations (actin:IRSp53 = 1:10) resulted in the number of actin seeds being significantly increased (median values: 0.0007 pixel^2^ [control], 0.001 pixel^2^ [0.25 μM IRSp53, i.e., actin:IRSp53 = 2:1], 0.0002 pixel^2^ [5 μM IRSp53, i.e., actin:IRSp53 = 1:10]). We used the same data as a control in graphs; however, to make the effect of 0.25 μM IRSp53 properly visible, the resolution of the scale was changed. The scale bar corresponds to 10 μm.

**Figure 11 ijms-24-13112-f011:**
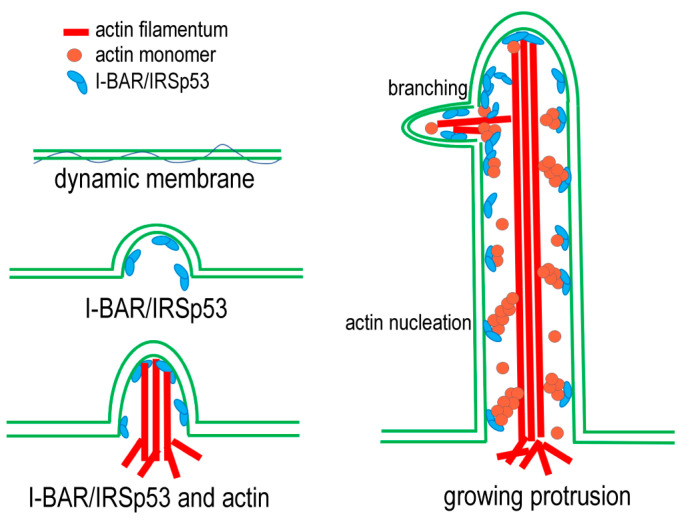
The schematic representation of the model describing the proposed function of IRSp53 and its N-terminal I-BAR domain in the formation of NTs. I-BAR domain proteins (associated or not with actin) can bind the dynamic cell membrane and change the organisation of lipids to promote membrane curvature. Next, the tubular structure is filled with F-actin. Actin filaments form bundles to improve the mechanical properties of the structure. In addition, both I-BAR and IRSp53 can bind to actin and thus influence the assembly of actin filaments by supporting the formation of actin nuclei and stabilising actin foci, resulting in more and longer tubes.

**Table 2 ijms-24-13112-t002:** The results of the analyses obtained with A20 B cells and their transgenic modifications.

Parameter	A20 Control	I-BAR Overexpression	IRSp53 Overexpression
Number of filopodia per cell	4.33 ± 0.51	16.30 ± 2.12	13.86 ± 1.65
Average length of filopodia (µm)	1.24 ± 0.06	3.05 ± 0.16	2.18 ± 0.11
Relative frequency of NTs	0.16 ± 0.03	0.12 ± 0.03	0.19 ± 0.04
Average diameter of the NTs (µm)	1.35 ± 0.10	0.93 ± 0.12	1.24 ±0.08
Length of the NTs (µm)	26.33 ± 5.59	21.36 ± 2.29	31.75 ± 4.17
Relative branching	0.12 ± 0.80	0.55 ± 0.19	1.00 ± 0.31

## Data Availability

The manuscript contains all the data that were generated within this study.

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
