# Peer review of "Molecular Relay Stations in Membrane Nanotubes: IRSp53 Involved in Actin-Based Force Generation"

_ijms, 2023, doi:10.3390/ijms241713112_

Round 1
Reviewer 1 Report
In this experimental study the authors investigated the role
of IRSp53 (BAIAP2 gene) in actin-based force generation in
membrane nanotubes of cell cutlure systems (COS-7 and A20 B).
Their findings suggest a complex multimolecular network of which
IRSp53 is an essential molecule to regulate the actin polymerization
dynamics of membrane nanotubes. Since these nanotubes networks are
important in cell-to-cell communication in health and disease,
the authors findings may have therapeutic implications. The
authors used superresolution microscopy TIRF (Total Internal
Reflection Fluorescent) configuration microscope to observe
and measure nanotubes response.
Concerns
1. The authors should refer to the previous application of
superresolution microscopy for filopodia observation and
localization of IRSp53 and its I-BAR domain and actin,
by Sudhaharan et al 2019 (doi: 10.1038/s41598-019-38851-w).
In Sudhaharan et al 2019 the quantitative interpretation of
filopodia using superresolution microscopy was attempted for
the first time.
2. Biochemical evaluation of the presented interactions
would add significantly in support of the proposed mechanism.
Reviewer 2 Report
In this manuscript Madarász et al. have studied the role of IRSp53 and its N-terminal I-BAR domain in the formation of cellular nanotubes (NTs). They found that both COS-7 kidney cells and A20 B cells express IRSp53, and that this protein co-localizes with actin in NTs. Moreover, they observed that overexpression of either IRSp53 or the I-BAR domain promoted cell morphological changes, giving to increased filopodia and NTs formation. The authors also show that IRSp53 and the I-BAR domain regulate actin nucleation/polymerization in a concentration-dependent manner, suggesting that plasma membrane-associated IRSp53 may be involved in NTs formation and elongation.
Major comment
The study by Madarász et al. is novel and interesting. The experimental approaches have been well performed and appropriated for this kind of investigations. However, some additional issues should be addressed before the manuscript can be considered for publication.
Specific points:
1.- Figure 1: The study of the expression of endogenous IRSp53 in COS-7 and A20 cells also requires Western blot analysis.
2.- Figures 2 and 3: To better show the subcellular localization of actin and IRSp53 in cells, a lower magnification of photomicrographs should be included together with their corresponding DIC images.
3.- How functional are the NTs that the authors have analysed in this work? The intercellular transport of particles, organelles or proteins should be studied by live cell video microscopy approaches. Results from these experiments would help to clarify the true nature of such structures, considerably increasing the reliability and strength of the study.
4. Why is NT length higher in cells overexpressing IRSp53 than in cells overexpressing the I-BAR domain? This result suggest that domains others than the I-BAR would be involved in NTs growing. Which could be the IRSp53 determinants responsible of NT enlargement? In searching for functional determinants, the protein structure of IRSp53 may be analysed beyond the I-BAR domain.
5.- Regarding the effects of IRSp53/I-BAR on actin dynamics, the authors’ finding that low concentrations increased actin polymerization whereas high concentrations decreased it, but increased actin nucleation is surprising. Which could be the physiological meaning of these findings? This set of results should be more convincingly discussed.
Round 2
Reviewer 1 Report
In this experimental study the authors investigated the role
of IRSp53 (BAIAP2 gene) in actin-based force generation in
membrane nanotubes of cell cutlure systems (COS-7 and A20 B).
Their findings suggest a complex multimolecular network of which
IRSp53 is an essential molecule to regulate the actin polymerization
dynamics of membrane nanotubes.
Reviewer 2 Report
The authors have satisfactorily addressed all my major concerns.
The manuscript has been improved.